# Application of Seaweed Generates Changes in the Substrate and Stimulates the Growth of Tomato Plants

**DOI:** 10.3390/plants12071520

**Published:** 2023-03-31

**Authors:** Adrian Alejandro Espinosa-Antón, Juan Francisco Zamora-Natera, Patricia Zarazúa-Villaseñor, Fernando Santacruz-Ruvalcaba, Carla Vanessa Sánchez-Hernández, Edith Águila Alcántara, Martha Isabel Torres-Morán, Ana Paulina Velasco-Ramírez, Rosalba Mireya Hernández-Herrera

**Affiliations:** 1Departamento de Botánica y Zoología, Centro Universitario de Ciencias Biológicas y Agropecuarias (CUCBA), Universidad de Guadalajara, Zapopan 45200, Mexico; 2Departamento de Desarrollo Rural Sustentable, Centro Universitario de Ciencias Biológicas y Agropecuarias (CUCBA), Universidad de Guadalajara, Zapopan 45200, Mexico; 3Departamento de Producción Agrícola, Centro Universitario de Ciencias Biológicas y Agropecuarias (CUCBA), Universidad de Guadalajara, Zapopan 45200, Mexico; 4Departamento de Agronomía, Facultad de Ciencias Agropecuarias, Universidad Central “Marta Abreu” de Las Villas, Santa Clara 54830, Cuba

**Keywords:** marine algae, physicochemical analysis, soil microbial, plant biostimulants, *Solanum lycopersicum*

## Abstract

*Ulva ohnoi* is a cosmopolitan green seaweed with commercial potential given the biomass that may be generated. We evaluated the effects of substrate changes induced by *U. ohnoi* application on the vegetative response of tomato plants under greenhouse conditions. First, the decomposition dynamics and N release of the dry seaweed biomass were studied using the litterbag method. Subsequently, we evaluated the effect of seaweed powder (SP) or seaweed extract (SE) applications on substrate and plant growth. Additionally, the growth parameters responses evaluated were related to the changes in substrate properties associated with each treatment. The results showed that the dry seaweed biomass has a rapid rate of degradation (k = 0.07 day^−1^) and N release (k = 0.024 day^−1^). The SP application improved the physicochemical and biological characteristics of the substrate by increasing the availability of minerals, the fungi:bacteria ratio, and the growth morphophysiological parameters (length, area, dry and fresh weight), chlorophyll and mineral content. In contrast, SE treatment showed a positive effect on the root, mineral content, and soil microbes. This study highlights the agricultural potential of *U. ohnoi* powder as an alternative supplement that supports nutrition and promotes the vegetative growth of plants cultivated in soilless horticultural systems.

## 1. Introduction

The development and implementation of new and innovative agricultural practices is urgently needed to counteract the negative effects of climate change, the continuous loss of arable land caused by overpopulation, and the deterioration of agroecosystems due to the excessive use of agrochemicals [1,2]. To this end, the use of plant biostimulants (PBs) may play a vital role in addressing the sustainability challenges of modern agriculture [3]. PBs are biological substances or microorganisms that stimulate the metabolic and physiological processes in plants responsible for efficient nutrient use, growth, productivity, and stress tolerance, regardless of whether or not the PBs contains beneficial nutrients [4,5].

In agricultural and horticultural crops, there is a growing interest in using cultivation methods such as natural biostimulants that influenced vegetative growth and improve the yield of fruits without any negative effects on plant quality. In addition, such a strategy allows for increasing biomass production as well as improves the nutraceutical quality of plant food [6]. The application of a plant biostimulant based on seaweed extract increased tomato fruit nutritional quality [7], and consequently enhanced the bioactive compounds in plant food matrices [8,9]. Seaweed extract application enhanced nutritional quality through direct plant provision of both macro- and micronutrients [4].

Plant substrates are materials that ensure adequate aeration, water, and nutrient supply while providing environments that are initially free of plant pathogens, unwanted seeds, and weeds [1,10,11]. The addition of organic materials to substrates to improve the performance of cultivated plants has produced favorable results [10,12]. In particular, the addition of seaweeds and seaweed products to plant substrates is quite promising given that these compounds are biodegradable, non-polluting, and non-toxic in nature [13,14]. In the past, seaweeds have been used as organic fertilizers in agricultural fields [14,15,16].

Recently, seaweed formulations have been used to enhance the production of different crops [17,18,19,20,21], as they provide notable amounts of mineral nutrients and organic compounds with plant biostimulant activity in the form of carbohydrates, amino acids, plant growth regulators, osmoprotectants, and antioxidants [2,14,22]. In addition, seaweeds contain various mucopolysaccharides, alditols, phenols, and organic materials that improve plant fertility and the retention of soil moisture [15,23].

Several efforts have been made to understand the action mechanisms of seaweed-based biostimulants [14,24], although most studies have generally focused on the direct effects on plants, while the possible impacts on soils or substrates have not yet received as much attention despite their great importance to crop performance [2,12]. This focus on plant effects can be attributed to the foliar application of seaweed-based biostimulants, which is the most common way of applying seaweed-derived products [16,22]. The incorporation of biostimulants into agricultural substrates in their natural forms (e.g., granules, fragments, or powders) represents a simple and low-cost means to achieve sustainable and economically and environmentally favorable horticultural outcomes [14,25,26].

Indeed, this approach is particularly promising for coastal farmers with easy access to macroalgae that may be incorporated into their cropping systems. However, knowledge of the decomposition and nutrient release rates of algal biomass is needed to reach informed decisions regarding their optimal use as PBs [13,27]. To date, studies that address this issue are scarce [28] or address the decomposition of fresh or dry algal biomass in aquatic ecosystems [29,30].

The green macroalgae *Ulva ohnoi* is an opportunistic species due to its rapid growth and the ecological plasticity it shows under a wide range of environmental conditions [31]. In eutrophic waters, *U. ohnoi* can accumulate in massive quantities to form green tides, which are well known due to their negative economic and environmental impacts on coastal communities [32]. The large amount of biomass deposited on coasts during green tides represents a highly exploitable resource for local communities; however, this biomass is often treated as rubbish and is not utilized [26,33]. *Ulva ohnoi* has also been successfully grown under different culture conditions and in intensive systems, which has resulted in the stable and commercial production of biomass [34,35]. However, few studies have evaluated the agricultural potential of this species [36,37,38].

The tomato (*Solanum lycopersicum* L.) is a member of the Solanaceae family and is one of the most important horticultural crops worldwide due to its nutritional, nutraceutical, and economic value [39,40]. Currently, the main challenges facing tomato production include optimizing the development of the vegetative and reproductive parts of tomato plants as well as yield in terms of productivity and fruit quality to satisfy the market demand while minimizing chemical inputs [20,23,41,42,43].

Thus, the aim of this study was to evaluate the effects of the application of powdered and aqueous *U. ohnoi* extracts on substrate properties and the vegetative response of tomato plants under greenhouse conditions. We hypothesized that the application of powdered and aqueous *U. ohnoi* extracts would result in different physical, chemical, and biological characteristics of the tomato growth substrate, which would be reflected in the growth, nutrient uptake efficiency, and assimilation of tomato plants.

## 2. Results and Discussion

### 2.1. C, N, and Cell Wall Composition of U. ohnoi

First, the fiber fraction composition of the *U. ohnoi* biomass was characterized. The majority of the fiber fractions were composed of cellular content (70.89%); Table 1), while 29.11% was neutral detergent fiber (NDF), which included cellulose (11.39%), hemicellulose (11.07%), and lignin (6.64%).

These results agree with those obtained by Castro-González et al. [44] and Yaich et al. [45], who reported a lower percentage of lignin compared to those of cellulose and hemicellulose in the insoluble fiber of *Ulva lactuca*. A previous proximate analyses of *U. ohnoi* described the richness of its soluble fibers, which were composed mainly of sulfated polysaccharides [34,37].

Most of the algal nutrients were in organic forms and needed to be mineralized to become plant-available. We determined the initial content of C (29.92%), N (3.12%), and the C:N ratio (9.59) of the dry biomass of *U. ohnoi* (Table 1) given the influence of these variables on the decomposition and mineralization rates of organic residues [27,46,47]. We found that the total N in *U. ohnoi* was higher than the N content reported by Cole et al. [36] for this species in land-based cultivation systems and lower than the values described by Magnusson et al. [37]. In addition, the C:N ratio agreed with the range of values reported in the literature for this species and those of other green seaweeds (5.7–11.3) [16,37,48]. The low C:N ratio of *U. ohnoi* suggests that it decomposes faster than other types of organic manure in soil [49,50,51,52,53].

### 2.2. Algae Decomposition and N Mineralization

The *U. ohnoi* decomposition experiment revealed an initial phase (0 to 14 d) in which the rapid loss of dry algal biomass (80%) was observed, followed by a second phase (15 to 56 d) in which the degradation of the algal biomass was relatively slower. The highest loss in *U. ohnoi* biomass (61% dry weight) occurred from 0 to 7 d. By day 15, the dry algae biomass had lost approximately 73% of its initial DW. Overall, 93% of *U. ohnoi* dry biomass was decomposed by day 56 (Figure 1a). Additionally, in the first week, 20% of the initial N was mineralized. This was followed by a period (8 to 35 d) of relatively constant N release. During the last week, the high release (approximately 25%) of the remaining N was recorded. By day 32, 50% of the initial N content in the dry algal biomass was mineralized. By the end of the experiment, approximately 70% of the total N of *U. ohnoi* was transferred to the substrate as a result of microbial activity (Figure 1b and Appendix A).

The loss of more than 50% dry weight of the algal material in the first week is consistent with reports from several authors who observed that this initial phase is characterized by the rapid degradation of hydrosoluble compounds, high microbial activity, and nutrient leaching/release [51,53]. The results in this study may be primarily associated with the high N content and low C:N ratio of the *U. ohnoi* biomass (Table 1), which met the metabolic and growth needs of the decomposer microorganisms [27,46,49]. In addition, *U. ohnoi* biomass contains relatively high concentrations of labile compounds (e.g., cellular content and soluble fiber) and readily degradable structural compounds (e.g., cellulose and hemicellulose) that may have been rapidly utilized by microorganisms [46].

From weeks 2 to 3, the loss of dry weight was relatively gradual and remained comparatively stable from week 4 until the end of the experiment. This was probably due to the accumulation of recalcitrant components present in the algal cell wall that were resistant to microbial activity [54,55]. Gupta and Singh [56] demonstrated that cell wall chemical content is an important predictor of the dynamics of organic matter (OM) decomposition. The cell wall polysaccharides of *Ulva* macroalgae constitute 38–54% of their dry biomass and include four families of polysaccharides: water-soluble ulvan and insoluble cellulose (major families) and peculiar alkali-soluble linear xyloglucan and a glucuronan (minor families) [57,58]. Ulvans are likely less degradable than the polysaccharides found in plant resources [13].

The relative decomposition rate of *U. ohnoi* under the experimental conditions in this study was 0.07 day^−1^ (Table 2). This value represents the speed at which organic materials decompose [23,50] and can fluctuate between 0.1 day^−1^ (rapidly decomposing residues) to 0.00001 day^−1^ (slowly degrading residues) [59]. According to this criterion, the dry algal biomass in our experimental conditions exhibited a fast rate of decomposition. The half-life time (t_½_) of *U. ohnoi* (defined as the time required for 50% of the dry mass to decompose) was approximately five days (Table 2).

The decomposition of *U. ohnoi* released approximately 70% of the N in the starting material with a rapid mineralization rate of 0.024 day^−1^ (Table 2).

These results support the use of the dry *U. ohnoi* biomass as a slow-release fertilizer that provides available forms of N for plant nutrition [28]. Other studies of different green manures applied to soil have reported rapid N mineralization when N content in the primary material is high (%N > 2 or C:N < 20) [52,60]. Likewise, the results obtained in this study may be related to the labile composition of the algae, which was characterized by low lignin values and lignin: N ratio (Table 1). These two factors are determinants of the net N mobilization/immobilization rates during OM transformation [27,49,61,62]. Finally, due to the rapid degradation and mineralization of the *U. ohnoi* dry biomass (Figure 1 and Table 2), we determined that the application of *U. ohnoi* powder on greenhouse tomato plants would take place seven days after transplanting following the methodology of previous studies by [18,19] and would continue at 2-week intervals. We are the first to report the decomposition dynamics and N mineralization over time of the dry biomass of a seaweed species under terrestrial conditions.

### 2.3. Physicochemical Analysis of U. ohnoi

The physicochemical analysis (Table 3) revealed the presence of chemical elements in the seaweed powder (SP) and seaweed extract (SE) that are required for the optimal growth and development of plants [42,63]. In this study, the pH values of SP (6.56) and SE (5.67) were slightly and moderately acidic, respectively. The electrical conductivity (EC) at 1:10 dilution (*w*/*v*) was 1.8 units higher in SP (3.78) than in SE (1.94) due to the higher concentration of mineral salts and cations in the former with respect to the latter. Higher pH (7.19) and EC (7.15 dS m^−1^) values than those found in this study have been reported in *U. ohnoi* cultivated in high-nutrient wastewater from shrimp farming [36]. Additionally, liquid extracts of *Ulva lactuca* and *U. flexuosa* [18,64] have shown similar EC values to those of this study, while pH was neutral.

The OM content of SP and SE was 71.26 and 0.024%, respectively. In addition, the total mineral content (ash) in the SP (28.74%) was higher than that reported for this macroalgae by Magnusson [37], and lower than the values reported by Cole et al. [36] and Mata et al. [34]. The protein content of SP (14.72%) and SE (0.026%) was estimated from the algae-specific nitrogen-to-protein conversion factor for green seaweed.

The protein fraction of *U. ohnoi* is rich in essential and non-essential amino acids [34,65], which are released by the hydrolysis (chemical or enzymatic) of these macromolecules and constitute an essential group of PBs [3,4,66].

Seaweed products are known to contain various minerals because seaweeds bioaccumulate the minerals found in seawater [39]. The predominant macronutrients in the SP and SE were N, K, and Mg and K and Mg, respectively, while the P concentration was low in both SP and SE. Moreover, Na, Fe, and Zn were the predominant micronutrients in SE and SP. Overall, these results agree with the macronutrient profile described for *U. ohnoi* biomass from land-based culture studies [36,37] and those of other *Ulva* species grown in natural environments [40,67]. Previous studies have proposed the use of *Ulva* species as liquid extract [19,40,67] or powder [37,68,69] biostimulants and biofertilizers for many agronomically important crops due to their rich content of inorganic nutrients.

### 2.4. Growth Promotion of Tomato Plant by SP and SE

In the greenhouse experiment, the SP and SE significantly (*p* < 0.05) promoted the growth of tomato plants. SP application significantly increased the values of all morphological variables evaluated. In comparison to those of the control plants, shoot and root length, leaf area, and root area increased by 25%, 40%, and 32%, respectively, while fresh shoot and root weight increased by 63 and 79%, respectively, and dry shoot and root weight increased by 40 and 60%, respectively.

We also observed a significant and beneficial effect of SE application at 0.2% on root length (25% increase), root area (14% increase), root fresh weight (39% increase), and root dry weight (31% increase) compared to what was observed with the control plants (Figure 2a–d).

The most pronounced stimulation was detected in plants grown on a SP-amended substrate. Several authors have shown that plants growing in soils treated with macroalgae manure or seaweed extracts exhibit a wide range of biostimulant effects during the vegetative phase that include stem elongation and increased leaf area, the stimulation of root cell division and lateral root/hair development, biomass accumulation, and an enhanced root-to-shoot ratio. In the majority of cases, the application of these compounds during the early vegetative phases of growth was found to be most effective [17,20,28,48,67].

Initially, the beneficial effects observed in plants were attributed to numerous mineral elements present in soluble forms in seaweed products with relevant functions during the vegetative phase [26,40,68]. Subsequently, seaweed biomass was found to contain various bioactive compounds that can directly act on plant metabolism and physiology [2,17,21,65,70] and modulate the synthesis and accumulation of endogenous metabolites involved in these processes [14,22].

In the present study, both SE and SP application to the substrate enhanced the root development and biomass accumulation when compared to what was observed in the control treatment. Figure 3 shows an example of a growth image pertaining to a representative plant of each treatment of *Ulva ohnoi*.

The significant and beneficial effects on root architecture could be due to the supply of macro- and micronutrients [18,19], the small levels of plant growth regulators present in the seaweed [17,71,72], and the promotion of the growth of beneficial bacteria that secrete root-promoting substances [23]. Seaweeds also have a positive effect on soil fertility by creating suitable environments for root growth [25].

As such, the significant increase in radical length, number of lateral roots, and fresh and dry weights of the root system after *U. lactuca* application in *Vicia faba* L. and *Capsicum annum* L. plants can be attributed to the effect of auxins and cytokinins present in algal extracts [17,72]. Additionally, the low zeatin concentrations in seaweed have been found to stimulate the growth of tomato roots [71]. Furthermore, it has been shown that algae sugars (polysaccharides or their derived oligosaccharides) can enhance plant growth in a manner similar to phytohormones [15,73]. Indeed, polysaccharide-enriched extracts from *U. lactuca* were found to increase the number of rooted *Vigna radiata* L. hypocotyl cuttings when compared to the effects of the indole-3-butyric acid group [68]. In *Zea mays* L., these polysaccharide-enriched extracts were found to induce large significant increases in radicle and root length and fresh and dry weight when compared to what was observed in the control treatment [21].

The formation of a vigorous root system following the application of seaweed compounds encourages morphophysiological changes in other vegetative organs [42]. However, SE application in this study did not significantly affect the growth parameters related to the aerial portion of the plants (Figure 3). These differential effects may be associated with the low OM content of the SE (Table 3). SE ashing leads to a loss of stimulatory effects, which has been confirmed by the role that the organic fraction plays in triggering beneficial growth responses in plants [71]. In addition, the adsorption of the organic and inorganic nutrients present in the SE by soil particles occurs often, which may reduce their instantaneous mobility and plant interactions [23]. On the contrary, the SP-treatment significantly enhanced the growth and development characteristics of aerial vegetative organs in terms of length, surface area, and biomass (Figure 3). Similar results due to seaweed-amended soil have been reported in tomato plants by Kumari et al. [25], Baroud et al. [20], and Sekhouna et al. [69]. These authors suggest that enhanced shoot growth could be related to enhanced nutrient use efficiency from the soil or substrate and enhanced photosynthetic capacity.

### 2.5. Mineral Composition and Chlorophyll Content

After 45 d, the effect of SP and SE application on the uptake and assimilation of inorganic elements in tomato plants was analyzed based on leaf mineral content. The mineral composition varied with the different treatments. The SP-treated plants showed a statistically significant increase (*p* < 0.05) in all inorganic nutrients compared to the control plants (Figure 4a,b). The total chlorophyll was measured using an SPAD-502 Plus portable chlorophyll meter (Spectrum Technologies, Ltd., Bridgend, UK). The chlorophyll content in the leaves of tomato plants grown on a SP-amended substrate significantly increased by 3.47 soil and plant analysis development (SPAD) units (*p* < 0.05) compared to that of the control plants. Similarly, the chlorophyll content of plants grown under SE treatment increased by 0.44 SPAD units over that of the control plants, although the differences were not significant (Figure 4c).

Plant metabolism and physiology depend on the continuous absorption of essential elements available in the form of different mineral compounds. These processes can vary as a function of the ionic species and the pH and EC of the substrate [1,55]. It has been shown that during the vegetative period, tomato plants require mainly N, followed by K, P, Ca, Mg, and S [37,38]. Previous studies agree with our results and have reported increases in macro- (N, P, K, Mg, and Ca) and micronutrient (Zn, Cu, Fe, and Mn) concentrations following *Ulva* spp. application in crops such as *Vicia faba* [17], *Zea mays* [28], *Solanum lycopersicum* [67], *Vigna radiata* [48], and *Raphanus sativus* [69]. This improvement in plant nutrient uptake due to the application of algae and algae derivatives may be attributed to the direct effects of their bioactive compounds on (i) root system development, (ii) root cell membrane permeability by the upregulation of genes encoding nutrient transporters, (iii) plant biochemical activity, and (iv) the development and physiological performance of leaves [14,22,74].

For instance, Mugnai et al. [73] applied a bioactive substance extracted from *Ulvales* green algae (EXT116) to the root of *Vitis vinifera* L. to evaluate its effect on NH_4_^+^ and K^+^ uptake. The results indicated that EXT116 significantly increased the influx of both ions in the root region spanning 0.8 to 1.7 mm from the root apex. In another study with *Brassica napus* L., plants treated with a commercial extract of the macroalgae *Ascophyllum nodosum* showed a significant increase in the acquisition of Cu and S. A transcriptional study revealed that this result was due to an overexpression of the *COPT2* and *BnSultr1.1/BnSultr1.2* genes, which encode root transporters associated with Cu and S uptake, respectively, and reported increased translocation of Fe and Zn from roots to shoots associated with the overexpression of the *NRAMP3* gene involved in this process [24].

The application of seaweed products in adequate quantities indirectly affects the nutrient acquisition capacity of plants by improving the physical, chemical, and biological characteristics of the soil or substrate [23]. *Ulva ohnoi* contains polyanionic compounds with strong chelating activity as phenolics [35] and ulvans [21,57] that can form complexes with metal ions essential for plant nutrition [15,16,22]. Likewise, some amino acids (e.g., cysteine, glycine, histidine, and glutamic acid) reported in the species [34,66] bind to some trace elements to form very small and electrically neutral chelates, which accelerate their uptake and transport within the plant [75]. Promoting the growth of root-associated beneficial microorganisms can also enhance nutrient cycling and delivery to plant roots [76]. In particular, the increased phosphorous content in the leaves of plants grown on the SP-amended substrate may have been due to the enhanced contribution of bioavailable forms or to fungal proliferation. Soil phosphorus usually limits plant growth due to its poor mobility [77]. However, some fungal taxa have the ability to produce extracellular enzymes to hydrolyze organic P and thus improve its efficient use by the plant [78].

In this study, plants treated with the SE (0.2%) significantly increased (*p* < 0.05) their content of K (14%), Na (27%), Fe (84%), Mn (46%), and Zn (15%) compared to the control plants, with the remaining minerals being assimilated in similar amounts, except for Ca. Cragie [15] reported that the quantitative aspect of plant nutrients cannot be very marked when extracts are used at low dilutions. This is perhaps related to a reduction in the instantaneous mobility of the inorganic nutrients of the SE due to substrate particle adsorption and their immobilization within microbial biomass due to the resulting stimulation of fungal and bacterial growth [14,23]. Finally, alkaline pH values of the substrate could reduce the solubility of some minerals and consequently their bioavailability in terms of plant uptake [55,69,74].

Chlorophyll content is an accurate indicator of the overall tissue development, plant stress levels, and CO_2_ uptake and is used to monitor N requirements during the crop cycle [35,75]. In previous studies, increases in chlorophyll content have been observed in tomato plants grown in seaweed-amended soils [20,25,69] or those treated with SEs [14,20,40,43,79,80]. However, SE application did not show any effect on chlorophyll content in this study. The stimulation of chlorophyll synthesis in tomato plants is likely due in part to an improvement in the absorption and assimilation of essential elements needed for pigment formation, such as N, Mg and Fe, whose content was also significantly higher in SP-treated plants compared to that of the control plants [42,74]. In addition, the possible presence of cytokinins and betaines that have been previously identified in *Ulva* species [17,70,72] may have promoted chloroplast biogenesis and increased the chlorophyll content due to an inhibition of their degradation rates [14,22,80]. In addition, Hamouda et al. [21] found that using soluble polysaccharides (5 and 10 mg mL^−1^) extracted from *U. lactuca* as priming agents for *Z. mays* L. seeds resulted in a significant increase in photosynthetic pigments at the vegetative stage.

### 2.6. Physicochemical Properties of the Plant-Growth Substrate

In the present study, a mixture of organic (peat moss and garden soil) and inorganic (pumice and perlite) materials were used as the substrate in the tomato plant pots. At the end of the experiment after day 45, the physicochemical characteristics of the substrate were analyzed. The effects of SP and SE addition are reported in Table 4. Both SP and SE significantly (*p* < 0.05) affected substrate properties.

Overall, the particle density (PD) and bulk density (BD) values in the three groups (SP, SE, and Control,) agree with the desired range of values for a given substrate (PD = 1.45–2.65 g cm^−3^ and BD < 0.75 g cm^−3^; Table 4). There was no difference in the PD between the SP-amended substrate and the control, whereas the SE-treated substrate showed lower PD values than those of the control. The BD value was low in the SP-amended substrate, but higher in the SE-treated substrate compared to that of the control. On the other hand, the total porosity in the SP-substrate was higher (total pore space [TPS] = 72.36%) than in the SE-treated and control substrates (TPS = 66.32 and 68.95%, respectively; Table 4).

According to Abad-Berjon et al. [81,82], an ideal plant substrate consists of two or more components (organic or inorganic) that are mixed to ensure adequate aeration and the supply of water and nutrients to the receiving crop. A substrate should provide a sufficiently solid structure to produce a suitable balance of air and water for optimal plant development. These ideal conditions can be evaluated by physical properties such as PD, BD, and total pore space [1,10]. Overall, the PD and BD values in the three experimental groups in this study fall within the desired ranges established by Abad-Berjon et al. [81]. In addition, the total porosity value of the SP-amended substrate was favorable for crop development (≥70%) according to Cabrera [12].

In a previous study, biomass of the brown alga *Cystoseira baccata* was used as a component of plant substrates and was found to reduce BD and increase porosity [10]. Decreasing substrate compaction by adding seaweed promotes plant root development and thus benefits crop growth and yield due to improved water and nutrient uptake [11,81]. In general, the effects on soil physical fertility have been mainly attributed to the rheological characteristics of macroalgae cell wall polysaccharides (e.g., ulvans) such as viscosity, gelling, and ion binding [23,57]. In this study, *U. ohnoi* application influenced the chemical properties of the substrate. The pH values of the SP- and SE-treated substrates were 7.83 and 8.82, respectively, while that of the control substrate was higher (pH = 8.73; Table 4). According to Benton [41], all three of these pH values are higher than the optimal pH range for tomato plants (5.5 to 6.8). The dissolved salts of the SE (mainly Na^+^) directly increased the pH value of the substrate. The observed decrease in pH with SP addition could be due to the breakdown of its labile organic fraction and the subsequent release of organic acids. Similarly, Ahmed et al. [69] found a decrease in soil pH (pH = 7.2) amended with *U. lactuca*.

Electrical conductivity increased following SP addition (1.21 dS m^−1^) when compared with that of the control substrate (0.21 dS m^−1^) due to the alkaline-earth cation salts (Mg^2+^ and Ca^2+^) contained in the algal biomass (Table 4). However, a gene expression analysis of *SlHB7* (*Solanum lycopersicum HOMEOBOX 7*), a gene related to the saline stress response in tomatoes [83], confirmed that plants grown on a SP-amended substrate were not affected by increased EC, as they showed similar transcript levels as those of the control and SE-treated plants (Appendix A). Additionally, the EC values in this study were lower than the maximum recommended threshold (2.5 dS m^−1^) for tomato crops [41].

The total mineral content (ash) in the SP treatment was higher (by about 1 unit) than that of the control group, but lower than that of the SE treatment (Table 4). Not all nutrients released from algal derivates are available to crops due to multiple factors such as their potential loss due to denitrification or leaching, immobilization by microbial communities, or their incorporation into the recalcitrant fraction of soil organic matter [28,42,50].

Seaweed ingredients include macro- and microelement nutrients. The application of both the SP or SE to the substrate resulted in significant increases in available P (31 and 3%), K (46 and 10%), Na (47 and 19%), Ca (7 and 3%), and Mg (41 and 24%). In addition, the available Cu was only higher in the SP-treated substrate (by 25%) compared to that of the control substrate. In the SE-treated substrate, the concentrations of available Mn and Fe were higher (by 12 and 7%) compared to those of the control (Table 4). As previously noted, one advantage of the incorporation of whole seaweed into a substrate is that this method offers a long-term supply of mineral elements to the soil [10,81]. Although seaweed has a modest content of some inorganic nutrients, it may be useful when natural sources are lacking or as an alternative to the excessive use of chemical fertilizers [10,23]. In addition, seaweeds provide a complementary supply of nutrients that are usually deficient during the crop cycle [28,43], as they contain chelating compounds that can increase nutrient availability and retention [16,22,23]. The OM and organic carbon (OC) content values in the SE-treated substrate were 3.6 and 2.1 units higher than those of the control (Table 4). This result relates to the increase in the microbial biomass of the plant growth substrate. After their deaths, microorganisms contribute to soil fertility, as their residues make up to 30–50% of the OM present [76]. In turn, the SP-amended substrate showed lower OM content and similar OC content compared to that of the control substrate. In this regard, the OM contribution of *U. ohnoi* biomass to the substrate over the long-term is limited due to its rapid mineralization by microorganisms [84].

### 2.7. Microbial Populations in the Plant-Growth Substrate

Under natural conditions, plant roots are in continuous contact with soil microbial communities. These microbial interactions enhance growth, improve nutrient acquisition, promote stress resistance, and facilitate disease suppression [2,23]. Nevertheless, these interactions are often not considered in the agronomic research of crops grown on substrates [11]. The application of *U. ohnoi* to the substrate in this study affected the growth and structure of the indigenous microbial community after a 45-day experiment. The SE treatment significantly (*p* < 0.05) increased the total number of aerobic mesophilic bacteria and fungi (yeasts and molds) compared to those of the control treatment (by one and six times the number of colony forming units (CFU) g^−1^ in the substrate, respectively; Table 5).

A significant increase in total fungi (four times the number of CFU g^−1^ in the substrate) and not in bacteria was observed in the SP treatment (*p* < 0.05) compared to that of the control. In addition, the fungi to bacteria ratio (F:B) was determined as an indicator of microbial community structure in the substrate. Relative to the control, the F:B biomass ratio was 3.5 and 3.8 times higher in the SE- and SP-treated substrates, respectively (Table 5). Overall, these results indicate that *U. ohnoi* application (SP and SE) stimulated fungal growth to a greater extent than bacterial growth. This suggests that fungi played an important role in the degradation of the complex substances present in the algal biomass.

The predominance of fungi or bacteria depends on the quality of the organic residues used as soil amendments. The organic nutrients provided by the macroalgae, mainly in the form of carbohydrates, constitute an additional source of carbon and energy for heterotrophic microorganisms [16,23]. In this study, the higher cellulose content in the lignocellulosic fraction of *Ulva ohnoi* favored the proliferation of fungal populations because this fraction is an ideal food source for these microorganisms due to its high C:N ratio [85]. Likewise, seaweed-derived mineral nutrients function as cofactors of key enzymes that regulate microbial physiology and play relevant structural roles in various biomolecules [76]. We can also hypothesize that plants with root systems that are enhanced due to the application of seaweed-derived products supply more exudates, such as sugars and organic acids, to the root–soil interface that activate a plethora of microorganisms and thus benefit soil biogeochemical cycles [14,43].

Our results agree with previous studies that have reported the positive impacts of macroalgae on the soil microbial ecosystem. Wang et al. [85] applied 40 g kg^−1^ of products from *Lessonia nigrescens* and *L. flavicans* to the soil and obtained increases in the number of bacteria, fungi, and the F:B of 172%, 67%, and 150%, respectively, compared to those of the control. Moreover, these authors indicated that soil enzyme (e.g., invertase, urease, proteinase, and phosphatase) activities improved as the brown seaweed dose increased. Similarly, the application of alkaline extracts from the seaweeds *Durvillaea potatorum* and *A. nodosum* altered the microbiological processes of the soil by increasing the total bacterial count and amount of available N, which affected the diversity of the bacterial community by promoting the growth of some bacterial families related to soil health [43]. In general, the aforementioned effects of seaweeds on the rhizosphere or soil microbiome significantly influence plant growth and productivity (i.e., root, shoot, and fruit biomass). Thus, seaweed derivatives could potentially enhance the plant growth promotion traits of microbes associated with plant root systems [2]. Although these results are of great importance, additional studies are needed to deepen our understanding of the effects of applying seaweeds, such as *U. ohnoi*, to plant root systems with regard to the diversity and abundance of beneficial microbial groups and their specific interactions with plants during the crop cycle.

### 2.8. Principal Component Analysis

In addition to the supply of nutrients due to the addition of algae, the physicochemical conditions of the growth substrate can benefit soil microbes and consequently improve plant performance in soilless growing systems [11]. A biplot analysis and dendrogram classification were used to confirm the relationships expressed among the treatments (SP, SE, and control), the physicochemical composition of the substrate with the presence of soil microbes (bacteria and fungi), growth parameters (shoot and root length, leaf and root area, and fresh and dry weight), and biochemical composition (macro and micronutrients and chlorophyll content) of tomato plants.

The principal component analysis (PCA; Figure 5) revealed that two factors explained 90.38% of the total variance. Factor 1 (PC1) explained 65.28% of the variance and was positively correlated with root length and area, fresh shoot weight, leaf mineral content (N, S, Na, Mg, and Zn), and the minerals (P, K, Na, Ca, M) available in the substrate, and was negatively correlated with BD, pH, and Mg in the substrate. Factor 2 (PC2) explained 31.5% of the variance and was positively correlated with OM, OC, Fe, and Mn available in the substrate, shoot length, and bacteria and fungi counts, and was negatively correlated with TPS, ash, and available Zn in the substrate (Figure 5).

By plotting data according to PC1 and PC2, two clusters were identified that showed a clear separation among the substrate properties from the different treatments. The first group was composed of the SE-treated substrate that presented a high accumulation of OM, OC, available Fe and Mn, pH, and BD values as well as high bacteria and fungi counts. The second group was composed of the SP-amended substrate that showed a higher F:B ratio, ash accumulation, and available K, P, Ca, Na, Mg, and Cu content as well as high EC and TPS values (Figure 5).

The results indicated that the beneficial effect of *U. ohnoi* SP application on the vegetative response of tomato plants is related to physicochemical changes in the substrate and microbiological characteristics. Thus, the initial hypothesis of this study was supported. Van Gerrewey et al. [11] indicate that the characteristics of the growing substrate can beneficially impact specific microbes and enhance healthy and productive plant growth. Algal chemical compounds have been found to improve the water-holding capacity, aggregate stability, and aeration of soils, which can stimulate the growth of plant root systems, boost biological soil mineralization, and improve macro- and micromineral availability and plant uptake [16,23].

## 3. Materials and Methods

### 3.1. Algae Material and Preparation of the Seaweed Extract

*Ulva ohnoi* biomass was obtained from the land-based ponds of the commercial culture system of Company Marine Products of the Las Californias S. de R.L. de C.V., located in Ensenada, Baja California, Mexico. The species was identified by morphological and molecular methods [35].

The SE was produced in the Biotechnology Research Laboratory of the University of Guadalajara (Guadalajara, Mexico) following the methodology of Hernández-Herrera et al. [18]. Briefly, 2 g of dry algae powder was added to 1 L of distilled water and autoclaved at 121 °C for 15 min at 1.5 kg cm^−2^. The hot extract was filtered through Whatman No. 40 filter paper from Sigma-Aldrich (Merck KGaA, Darmstadt, Germany) and stored at −4 °C until further use. The SE was designated as a stock solution at a 0.2% concentration.

The SE and SP were analyzed in the Environmental Laboratory of Organic Fertilizers of University Center for Biological and Agricultural Sciences (CUCBA), in University of Guadalajara. The neutral and acid detergent fiber (NDF and ADF, respectively) and acid detergent lignin (ADL) fractions of the algal biomass were determined [86]. The cellular content of the fiber fractions was determined by subtracting the quantity of cell walls (NDF) from 100 [40]. The amount of hemicellulose was calculated from the difference between NDF and ADF, and the amount of cellulose was obtained by subtracting ADL from ADF [47]. Both EC and pH were measured using a YSI 35 Conductance Meter (Yellow Springs Instruments Co., Yellow Springs, OH, USA) or HI 2211 pH Meter (Hanna Instruments^®^, Mexico City, Mexico) with a 1:10 (*w*/*v*) dilution.

Finally, the chemical analyses were performed following the Official Methods of Analysis of the Association of Official Analytical Chemists [87]. The ash (dry inorganic) content was determined by calcination at 550 °C for 6 h in a MA12D muffle (method 942.05), and OM content was calculated based on the difference with respect to 100% ash. Nitrogen content was determined by the micro-Kjeldahl method (method 976.05). Protein content was indirectly estimated (method 954.04) using a protein conversion factor of 4.72 [88]. Total carbon content was determined by dry combustion in a TruSpec^®^ CHNS-O elemental analyzer (LECO Corporation, St. Joseph, MI, USA). The content of K, Na, Ca, Mg, Cu, Mn, Zn, and Fe was analyzed by atomic absorption spectrophotometry, and phosphorus content was determined by colorimetry. 

### 3.2. Algae Decomposition and Nitrogen Mineralization

The decomposition dynamics of *U. ohnoi* in soil were analyzed using the litterbag method described by Wider and Lang [89]. Briefly, a known mass (5 ± 0.0001 g) of fragmented dried seaweed was enclosed in a 5 cm × 10 cm (W × L) litterbag with 1.0 mm openings (Ankom Scientific, New York, NY, USA). The mesh size allowed microbes and micro- and mesofauna access to the soil while preventing the loss of algal biomass [90]. Three bags of leaf litter were placed 5 cm deep into each pot (30.5 cm × 36 cm, width × length), which contained a mixture of vermiculite (Termolita S.A., Santa Catarina, NL, Mexico), peat moss (Sunshine Mix 3™), pumice, and garden soil in a ratio of 1:1:1:1 (*v*/*v*; Figure 6a).

The decomposition experiment was conducted in a greenhouse (Figure 6b). For this, the irrigation frequency was determined by environmental conditions, and the moisture within pots was maintained at 80% capacity. On each sampling day, the three leaf litter bags were collected from each designated pot. The sampling days were set at 7, 14, 21, 28, 35, 42, 49, and 56 d after the start of the experiment on 10 March 2022 (Figure 6c). Upon collection, all external material adhering to the litterbags was removed with a brush. After collection, the litterbag content was dried in a forced air circulation oven (10–180, Quincy lab, Inc., Burr Ridge, IL, USA) at 70 °C for 72 h. After which, the materials were ground and passed through a 2 mm sieve. The samples were then weighed with an analytical balance (HR-200, A&D Company, Ltd., Ann Arbor, MI, USA), and mass loss was determined as the difference between the initial and final weights.

The total N content in the algal residues was determined in triplicate by the same method used to chemically characterize *U. ohnoi*. From these data, the rate of decomposition and N release from the *U. ohnoi* biomass over time was estimated using the single-component exponential model proposed by Olson [91]:Y = a exp(−kt)(1)
where y is the dry weight or N content remaining at time t (days), k is the decomposition or nutrient release constant (day^−1^), and the parameter a is the initial amount of dry mass or N content [27,54]. By linearizing this equation, the relative decomposition rate (k) can be calculated:k = ln (a/y)/t(2)

With the value of the decomposition constant, the half-life (t_½_), or the time (days) required for 50% of the dry mass of the algae to decompose or mineralize nitrogen, was calculated using the equation employed by [47,54]:t_½_ = ln (2)/k(3)

### 3.3. Plant Material and Growth Conditions

Tomato (*Solanum lycopersicum* L. cv. “Rio Fuego”) seeds (Kristen Seed, San Diego, CA, USA) were sown on peat moss media (Sunshine Mix 3^TM^) in seedling trays and allowed to germinate for 15 d. Thirty-six seedlings were transplanted into individual 1-L pots (11 cm × 15 cm, width × length) that contained a mixture of organic and inorganic materials described in Section 3.2 (Figure 6d). The growing substrate was prepared following the recommendations for the production of plants in containers [12,81]. The tomato plants were fertilized 1 week after being transplanted with 50 mL of 20:20:20 (N-P-K) soil drench solution (Peters Professional, Scotts-Sierra Horticultural Products Co., Marysville, CA, USA). Fertilization continued at 2-week intervals up to the end of the experiment. Each pot was supplied with a consistent volume of water each day to maintain a field moisture capacity of 75%. The tomato plants were grown for 45 d under natural light conditions. The temperature ranged from 16 ± 2 °C (night) to 30 ± 2 °C (day), with 70 to 85% relative humidity during the growth stage.

### 3.4. Application of U. ohnoi SP and SE

Tomato plants were cultivated in a greenhouse from April to June 2022 at CUCBA. The experiment began with 15-day-old plants and concluded with 45-day-old plants. The plants were arranged in a completely random design and assigned to three treatment groups consisting of 12 experimental unit plots. The experiment included three treatments: control plants (without algae), plants treated with 50 mL of the SE at 0.2%, and plants treated with 5 g of SP. Both the SE and SP were applied one week after transplanting and added directly on the surface of the substrate every 7 and 15 d, respectively. The dose and concentration of the SE were defined based on the results described by Hernández-Herrera et al. [18]. The application time of the SP (15 d) was determined from the results of *U. ohnoi* biomass mineralization in the decomposition experiment (see Section 2.2).

### 3.5. Morphological Atributes and Chlorophyll Content of the Tomato Plants

The effect of *U. ohnoi* application on the growth of tomato plants 45 d after transplanting (DAT) was evaluated. Leaf chlorophyll content was measured in all plants per treatment using the third composite leaf from the base. Three readings were taken from each leaf with a SPAD 502 Plus Chlorophyll Meter (Spectrum Technology, Inc., Aurora, IL, USA) and expressed as a ratio (equivalent to SPAD units) [79]. Morphological characteristics, such as shoot length, main root length, fresh weight, dry weight (g), and leaf and root area (cm^2^), were measured. For this, the plants were carefully removed from their pots and immediately submerged for 10 min in bowls filled with water. Then, the root system was carefully washed to remove substrate particles and subsequently dried in an oven (Terlab MA H45DM, Terlab, SA de CV, Zapopan, Mexico) at 60 °C for 72 h. All 12 plants per treatment were photographed, and their growth characteristics were measured using ImageJ v.1.52a software (https://imagej.nih.gov/ij/download.html, accessed on 4 July 2022). Immediately after, the plants were cut at the collar region and separated into the shoot (stem and leaf) and root portions to measure their fresh weights with a precision balance (BJ–410C, Precisa, Zurich, Switzerland). After the growth characteristics were measured, leaf samples were collected, placed in paper bags, and oven dried at 65 °C for 48 h before weighing and mineral analysis. The macronutrient (N, P, K, S, Mg, Ca, and Na) and micronutrient (Zn, Fe, Mn, and Cu) content were determined by the same method described in Section 3.1. In addition, to corroborate that the salt contained in the algae powder did not affect the plants growth, the SlHB7 gene expression related to the response to salinity stress in tomato was performed according to the procedure described by Becerril–Espinosa et al. [92].

### 3.6. Physicochemical Properties of the Substrate

The physicochemical analyses of the substrates from the SP, SE, and control treatments were carried out in the Environmental Laboratory of Organic Fertilizers of CUCBA. The substrate samples from each treatment were collected at 45 d at the end of the experiment. The substrate samples were collected from the surface, on the middle zone and the bottom of the pots (500 g of mixed subsamples) were divided into two parts. The first was used for the physicochemical evaluation, and the second was used for additional microbial analysis. Bulk density and PD were determined through the procedure described by Gabriels et al. [93], and TPS was determined following the methods of De Boodt et al. [94]. Ash and OM content were determined following the AOAC methods described in Section 3.1. To analyze the content of available minerals (Na, K, Ca, Mg, Cu, Mn, Zn, and Fe), the samples were subjected to acid digestion and analyzed by atomic absorption spectrophotometry [95], and the spectrophotometry method of Hoffman [96] was followed to determine P content. Total OC was determined according to the procedures described by Walkley and Black [97]. Both pH (H_2_O) and CE were measured in an aqueous medium with the substrate at a ratio of 1:10 *w*/*v* [98]. 

To estimate the total number of aerobic mesophilic bacteria and fungi (yeast and mold), 10 g of substrate were used following the plate count method of Onet et al. [99]. Briefly, plate count agar medium (pH 7 ± 0.2, incubation at 37 °C for 3 d) and potato dextrose agar medium (pH 5.6 ± 0.2, incubation at 25 ± 2 °C for 4–5 d) were used to culture the bacteria and fungi, respectively. The number of microorganisms present in the samples was quantified and expressed as CFU per gram of substrate (CFU g^−1^). In addition, the fungi-to-bacteria ratio was also calculated.

### 3.7. Statistical Analysis

All data from the decomposition and greenhouse growth experiments were evaluated for normality (Shapiro–Wilk test) and homoscedasticity (Levene test). For all experiments, a one-way analysis of variance (ANOVA) was used to compare treatments, and the Holm–Sidak post hoc multiple comparison test was used to evaluate the differences between means (*p* < 0.05). A joint principal component analysis (PCA) was conducted, and a biplot analysis was used to confirm the relationships that were expressed among the treatments (SP, SE, and control), the physicochemical composition of the treated substrates, the presence of soil microbes (bacteria and fungi), and the growth (shoot and root length, leaf and root area, and fresh and dry weight) and biochemical composition (macro and micronutrients and chlorophyll) of the tomato plants. In addition, a cluster/SIMPROF analysis was performed based on the Euclidian distance matrices constructed from descriptor data to identify similar patterns among the variables as well as the characteristics that had the highest descriptive values and that best explained the chemical composition of the treatments, the chemical composition of the substrate, the presence of soil microbes, and the morphological and biochemical variables of the tomato plants. The data were statistically processed as mean values using the statistical package Statgraphics Centurion XVI.II for Windows.

## 4. Conclusions

The application of powdered *U. ohnoi*, a green macroalgae, to the roots of tomato plants cultivated in pots was more effective in promoting vegetative growth than the application of an aqueous extract from the same species. The rapid decomposition of the powder in the growth substrate facilitated the release of macro- and micronutrients and other organic compounds, which had a dual effect on growth parameters (root and shoot length, area, fresh weight, dry weight, leaf mineral content, and chlorophyll). First, the bioactive metabolites present in the seaweed biomass likely directly benefited plant metabolism and physiology during the vegetative period. Second, the SP-amended substrate increased the availability of inorganic nutrients, and substrate porosity, all of which improved the root system development, nutrient uptake and assimilation, and chlorophyll synthesis, and consequently benefited the morphological attributes of the tomato plants. For instance, specific bacteria together with fungi may create a more indirect synergism that supports plant growth, capable of transforming and stabilizing inputs; and the fungi to bacteria ratio (F:B) ration biomass has a greater C:N ratio, which results in an increased carbon use efficiency, nutrient acquisition and enhancement of root branching. In addition, the fungi themselves have also been shown to have an impact on the composition of bacterial communities. However, microbial effects on plants can also be neutral or positive depending on the different microbial groups involved. Our results highlight the agricultural potential of *U. ohnoi* powder as an alternative supplement that supports nutrition and promotes the vegetative growth of plants cultivated in soilless horticultural systems. To the best of our knowledge, this is the first study to evaluate the biostimulant effects of *U. ohnoi* powder and its aqueous extract on tomato crops under soilless culture conditions.

## Figures and Tables

**Figure 1 plants-12-01520-f001:**
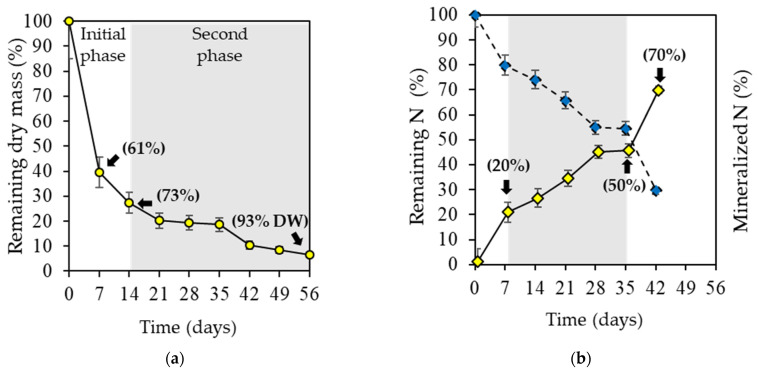
Decomposition and N mineralization of *Ulva ohnoi* over time after being incorporated into the substrate: (**a**) percentage of biomass remaining (dry weight [DW]) and (**b**) N remaining in the decomposed dry biomass (dotted line) and N released to the substrate (solid line). Values represent mean ± standard deviation (n = 3).

**Figure 2 plants-12-01520-f002:**
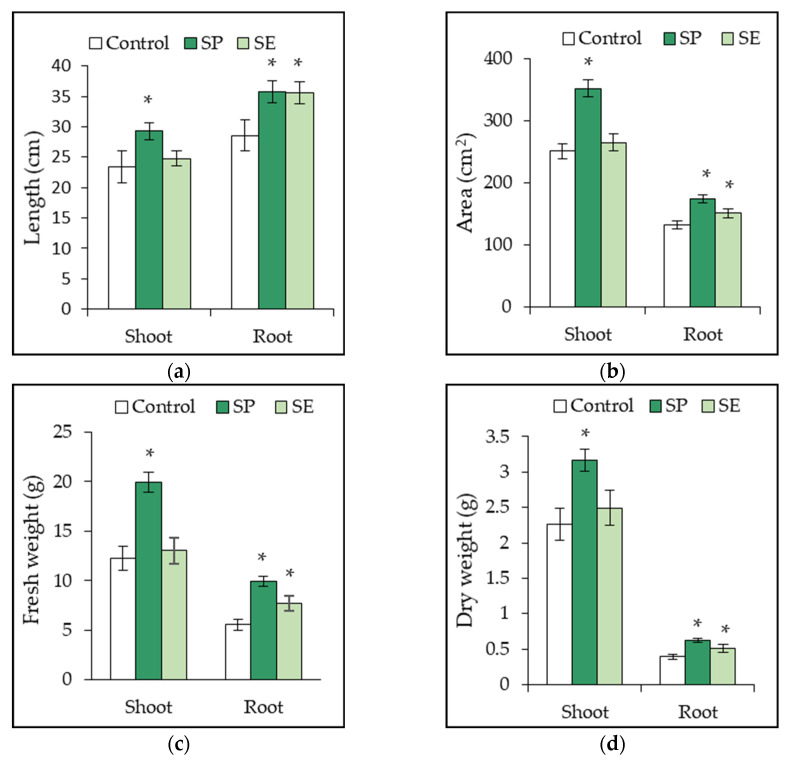
Morphological parameters of (**a**) length, (**b**) area, (**c**) fresh weight and (**d**) dry weight of 45-day-old tomato plants treated with seaweed powder (SP) and seaweed extract (SE) from *Ulva ohnoi*. Values represent mean ± standard deviation (n = 12 plants). Asterisks (*) over the bars indicate significant differences of each treatment in comparison to the control (untreated plants), based on the Holm–Sidak means comparison test (*p* < 0.05).

**Figure 3 plants-12-01520-f003:**
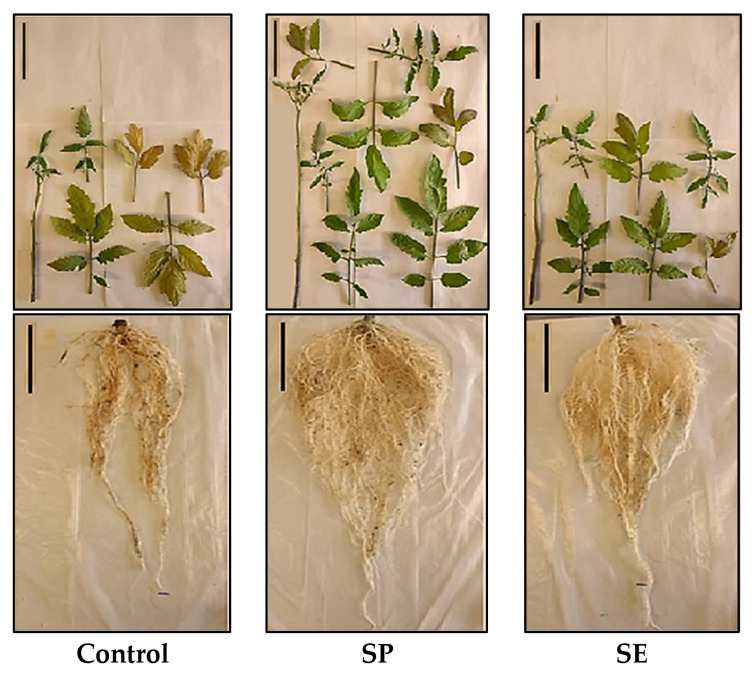
Tomato plant growth after the 45-day experiment. Treatments included control (untreated plants), plants treated with seaweed powder (SP) and plants irrigated with seaweed extract (SE) from *Ulva ohnoi*. Bar = 5 cm.

**Figure 4 plants-12-01520-f004:**
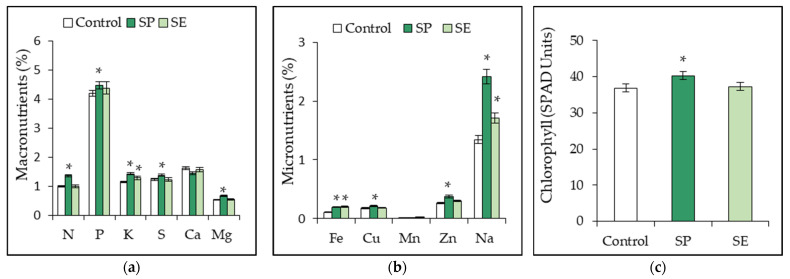
Mineral composition: (**a**) macronutrients, (**b**) micronutrients (% dry weight), and (**c**) chlorophyll content (soil and plant analysis development [SPAD] units) in the leaves of 45-day-old tomato plants treated with seaweed powder (SP) or seaweed extract (SE) from *Ulva ohnoi*. Values represent mean ± standard deviation (n = 12 for chlorophyll and n = 3 (mix of 4 plants each one) for macro- and micronutrients). Asterisks (*) over the bars indicate significant differences of each treatment in comparison to the control (untreated plants), based on the Holm–Sidak means comparison test (*p* < 0.05).

**Figure 5 plants-12-01520-f005:**
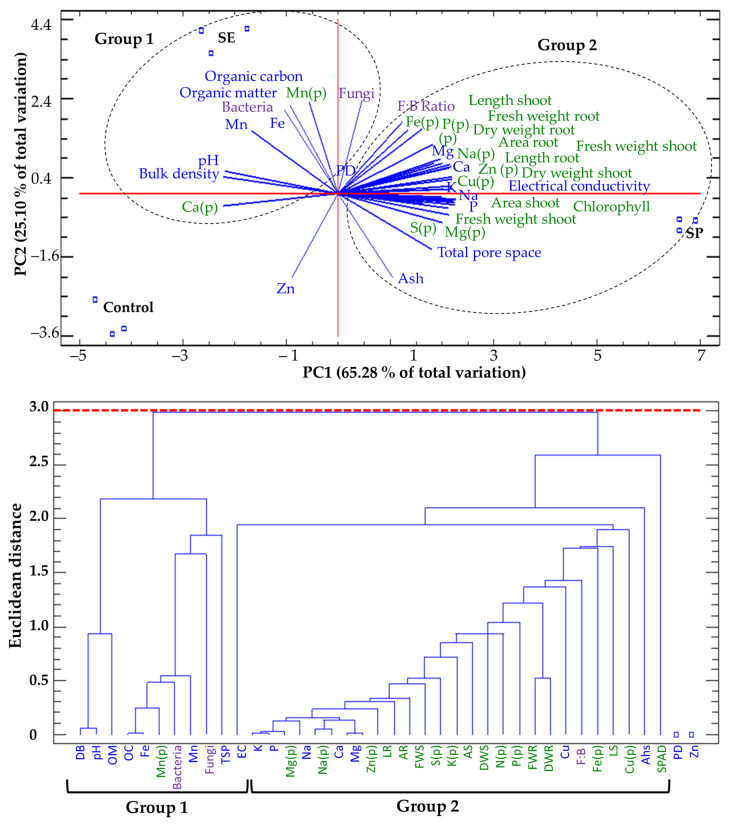
Principal component analysis (PCA) and dendrogram classification based on the physicochemical characteristics of substrates treated with seaweed powder (SP) or seaweed extract (SE) and the control substrate on soil microbes (fungi and bacteria) and the growth and physiological characteristics of tomato plants after a 45-day experiment. The physicochemical (blue letters), soil microbiological growth (purple letters), and morphological and biochemical (green letters) characteristics of the substrates used to grow tomato plants are shown.

**Figure 6 plants-12-01520-f006:**
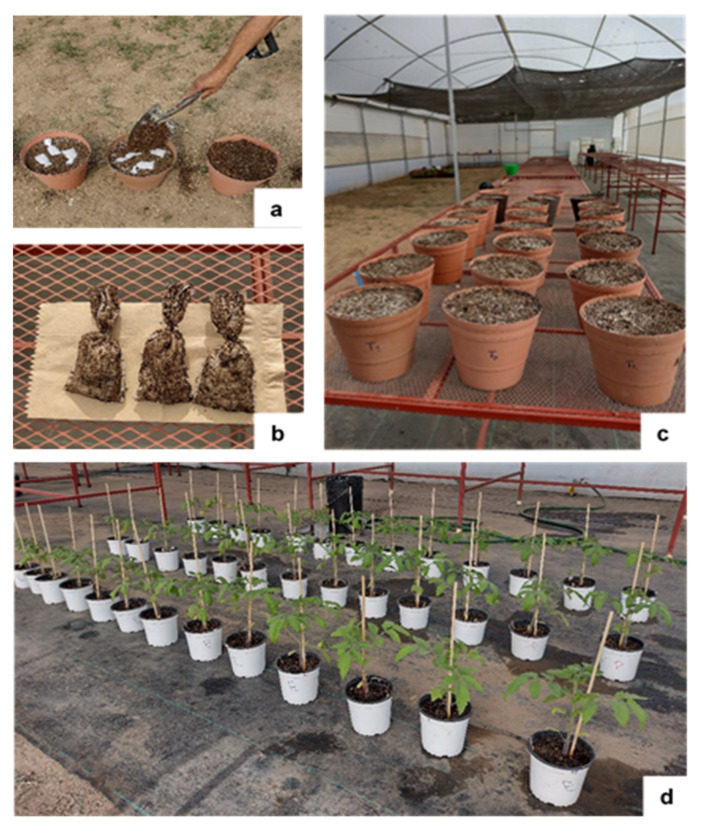
Decomposition in soil of *Ulva ohnoi*. (**a**) Three litter litterbags (1.0 mm openings) with fragmented dried seaweed were placed 5 cm deep into each pot, (**b**) the collected litterbag sampling days at 7, 14, 21, 28, 35, 42, 49, and 56 d after the start of the experiment, (**c**) image of the pots and the litterbag method in the greenhouse and (**d**) arrangement of experimental units in the plant growth at 45-days-old.

**Table 1 plants-12-01520-t001:** Fiber fraction composition and C and N content of *Ulva ohnoi* (% *w*/*w* on a dry basis).

Component	Value
Cellular content	70.89 ± 0.04
Neutral detergent fiber (NDF)	29.11 ± 0.02
Acid detergent fiber (ADF)	18.04 ± 0.43
Lignin	6.64 ± 0.01
Cellulose	11.39 ± 0.43
Hemicellulose	11.07 ± 0.45
Carbon	29.92 ± 0.13
Nitrogen	3.12 ± 0.04
C/N ratio	9.59 ± 0.08
Lignin/N ratio	1.94 ± 0.00

Values represent means ± standard deviation (n = 3).

**Table 2 plants-12-01520-t002:** Decomposition dynamic and N release of *Ulva ohnoi*.

Parameter	k (day^−1^)	t_½_ (days)
Dry weight loss *	0.07 ± 0.03	4.84 ± 1.5
Nitrogen release **	0.024 ± 0.01	31.74 ± 8.09

Values represent mean ± standard deviation corresponding to * n = 8 or ** n = 6. K = decomposition or nutrient release constant estimated from the single exponential model Y = a exp(−kt), t_½_ = half–life time.

**Table 3 plants-12-01520-t003:** Physicochemical characteristics of seaweed powder and seaweed extract from *Ulva ohnoi*.

Analysis	SP	SE
pH	6.56 ± 0.00	5.67 ± 0.00
Electrical conductivity (dS m^−1^)	3.78 ± 0.01	1.94 ± 0.00
Organic matter (%)	71.26 ± 0.02	0.14 ± 0.01
Protein (%)	14.72 ± 0.01	0.026 ± 0.00
Ash (%)	28.74 ± 0.02	nd
Macronutrients (%)		
C	29.92 ± 0.13	0.059 ± 0.00
N	3.12 ± 0.04	0.005 ± 0.00
P	0.09 ± 0.02	0.006 ± 0.00
K	3.98 ± 0.00	0.053 ± 0.00
Ca	0.49 ± 0.00	0.019 ± 0.00
Mg	1.83 ± 0.01	0.029 ± 0.00
Micronutrients (mg kg^−1^)		
Na	2560 ± 1.52	361 ± 1.50
Cu	12.70 ± 0.05	1.60 ± 0.01
Mn	4.80 ± 0.02	<1 ± 0.00
Fe	142 ± 0.57	12 ± 0.57
Zn	27 ± 0.12	27 ± 0.57

Values represent means ± standard deviation (n = 3). Treatments seaweed powder (SP) and seaweed extract (SE). nd = not determined.

**Table 4 plants-12-01520-t004:** Physicochemical characteristics of the substrate after a 45-day experiment.

Property	Control	SP	SE
Bulk density (g cm^−3^)	0.57 ± 0.001 ^b^	0.50 ± 0.017 ^c^	0.61 ± 0.005 ^a^
Particle density (g cm^−3^)	1.82 ± 0.005 ^a^	1.82 ± 0.004 ^a^	1.79 ± 0.007 ^b^
Total pore space (%)	68.95 ± 0.389 ^b^	72.36 ± 1.017 ^a^	66.32 ± 0.374 ^c^
pH	8.73 ± 0.015 ^b^	7.83 ± 0.01 ^c^	8.82 ± 0.006 ^a^
Electrical conductivity (dS m^−^^1^)	0.21 ± 0.015 ^c^	1.21 ± 0.006 ^a^	0.33 ± 0.01 ^b^
Organic matter (%)	16.84 ± 0.6 ^b^	16.13 ± 0.38 ^c^	20.42 ± 0.548 ^a^
Organic carbon (%)	9.77 ± 0.348 ^b^	9.36 ± 0.221 ^b^	11.84 ± 0.318 ^a^
Ash (%)	83.158 ± 0.6 ^b^	83.872 ± 0.38 ^a^	79.586 ± 0.548 ^c^
Availed minerals (mg kg^−1^)			
P	46.82 ± 0.388 ^c^	68.27 ± 0.484 ^a^	48.43 ± 0.755 ^b^
K	901.67 ± 0.306 ^c^	1663.80 ± 1.058 ^a^	1006.5 ± 0.624 ^b^
Na	757.00 ± 1.00 ^c^	1426.33 ± 1.155 ^a^	937.33 ± 1.528 ^b^
Ca	4815.67 ± 0.57 ^c^	5186.67 ± 1.528 ^a^	4990.33 ± 1.53 ^b^
Mg	1697.33 ± 0.58 ^c^	2892.33 ± 1.16 ^a^	2234.00 ± 1.00 ^b^
Cu	0.55 ± 0.012 ^b^	0.73 ± 0.015 ^a^	0.51 ± 0.015 ^c^
Mn	4.63 ± 0.01 ^b^	4.22 ± 0.006 ^c^	5.26 ± 0.071 ^a^
Fe	13.91 ± 0.107 ^b^	13.80 ± 0.015 ^c^	14.94 ± 0.071 ^a^
Zn	3.05 ± 0.064 ^a^	2.09 ± 0.02 ^b^	1.73 ± 0.036 ^c^

Treatments with seaweed powder (SP) and seaweed extract (SE) from *U. ohnoi*. Values represent mean ± standard deviation (n = 3). Different letters (a–c) within the rows indicate significant differences of each treatment in comparation to the control (untreated substrate) based on the Holm-Sidak means comparison test (*p* < 0.05).

**Table 5 plants-12-01520-t005:** Total number of microorganisms (colony forming units (CFU) g^−1^ in substrate dry weight (DW)) determined in substrates after a 45-day experiment.

Total Microorganisms (CFU × 10^3^ g^−1^)	Control	SP	SE
Bacteria	217.50 ± 12.62 ^b^	239.36 ± 43.07 ^b^	445.49 ± 6.51 ^a^
Fungi	110.27 ± 9.72 ^c^	456.00 ± 49.15 ^b^	769.98 ± 73.65 ^a^
Fungi/Bacteria ratio	0.51 ± 0.09 ^c^	1.92 ± 0.13 ^a^	1.73 ± 0.16 ^b^

Treatments included substrates treated with seaweed powder (SP) and seaweed extract (SE) from *U. ohnoi*. Values represent mean ± standard deviation (n = 3). Different letters (a–c) within the rows indicate significant differences of each treatment in comparison to the control (untreated substrate), based on the Holm–Sidak means comparison test (*p* < 0.05).

## Data Availability

Not applicable.

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
