# Peer review of "Application of Seaweed Generates Changes in the Substrate and Stimulates the Growth of Tomato Plants"

_plants, 2023, doi:10.3390/plants12071520_

Round 1
Reviewer 1 Report
According to methodology section. Abstract needs a little bit more of methodology.
For plant results. Authors in methodology say that they took pictures from plants but none is shown. Maybe for biostimulant effects it is helpful to see the growth of plants, shoot and root.
A question remainied open. In conclusión authors say that SP was better than SE but total number of microorganisms was better for SE. It has to do with fungi/bacteria ratio? Needs more discussion.
See comments in MS

Author Response
According to methodology section. Abstract needs a little bit more of methodology.
The abstract was changed because, the section is also too long. This section is changed to 200 words maximum.
For plant results. Authors in methodology say that they took pictures from plants but none is shown. Maybe for biostimulant effects it is helpful to see the growth of plants, shoot and root.
In line, 315. The author included the Figure 3. Tomato plant growth after the 45-day experiment.
A question remained open. In conclusion authors say that SP was better than SE but total number of microorganisms was better for SE. It has to do with fungi/bacteria ratio? Needs more discussion.
In Lines 755 to 761 the information was include in discussion.
See comments in MS
All changes made in the writing are showed in the PDF version, and are marked in blue for reviewer 1.

Reviewer 2 Report
The manuscript entitled “Substrate Application of Seaweed Powder from Ulvaohnoi Triggers Physicochemical and Microbial Changes and Stimulates the Growth of Tomato Plants”, authored by Espinosa-Antòn and colleagues, deals with the evaluation of the potential effects of substrate changes induced by U. ohnoi application on the vegetative response of tomato plants under greenhouse conditions.
The manuscript is written with authority and contains truly interesting data. However, before it can be judged suitable for publication in Plants something need to be fixed.
(i) The title is really too long and complicated. It is unusual to find title of such complex scientific articles. Strongly suggest authors to make this section shorter. Normally, a single sentence summarizing the research purposes is recommended for the Title.
(ii) In the affiliations section, authors should include acronyms for each component immediately after the respective e-mail address. These same acronyms should then be used in the contributions section.
(iii) The abstract section is also too long. Normally this section is 200 words maximum. Incidentally, this is also a prerogative of the journal. I would suggest reading the guidelines carefully before the next submission.
(iv) Some keywords should be changed. The utility of these terms is to facilitate the search of the article using common scientific search engines (PubMed, GoogleScholar, Scopus, etc.), which rely on the terms contained in title, abstract, and keywords. Consequently, using terms that are already in these sections as keywords is inappropriate. I strongly suggest that the repetitive keywords be changed before re-submission.
(v) The introduction contains a lack of information. In recent years, research is beginning to evaluate not only the effect of biostimulant application on plant physiology, but also on the quality of fruit that can be produced. This is a very important point, which has resulted in the shift of almost exclusive use of Arabidopsis in experimental trials. Tomato, finds an interesting application in these studies with biostimulants, precisely because the plant is capable of producing edible fruits. There are currently a limited number of scientific articles dealing with the potential effect of biostimulants on the nutritional, and nutraceutical properties of produced fruits. I strongly recommend that a paragraph in the introduction be introduced and relevant references included (10.3390/ijms232415911; 10.3390/agronomy12020428; 10.3390/plants11030348).
(vi) Table lack in standard deviation. Did the authors performed more than one determination for this analysis?
(vii) Figure 1 and 3 also lacks standard deviations. Each individual point should have a bar reporting the standard deviation. If the authors feel that the resulting graph is too confusing, they should at least introduce information in the caption reporting that the coefficient of variation is always under a certain percentage. I also recommend reporting these data as mean and standard deviation in a supplementary table, since it is not apparent whether the data are actually statistically significant.
(viii) table three should report an additional column with the P value.
(ix) what do SP and SE mean in the table? Their meaning should be explained as footnotes to the table.
(x) SPAD detection of chlorophylls is certainly a very fast and low-cost method for these determinations. However it turns out to be not very accurate. If the authors have the ability to perform quantifications spectrophotometrically or in HPLC of their degradation products it would be better.
(xi) Why is paragraph 2.4. called biochemical content? What do the authors mean by biochemical content? I don't think such a term is appropriate. Biochemistry studies biological macromolecules, how they function, and the processes in which they are involved. In this section I see none of that. The authors should rename this section as "Mineral composition and chlorophyll content" or something like that.
(xii) letters related to ANOVA, when present in tables, should be position as superscript of the last number.
(xiii) In the affiliation section, authors should list their email addresses for each affiliation, along with their acronyms in parentheses. These acronyms should be the same as those used for the contributions section.
Author Response
REVIEWER 2.
The manuscript entitled “Substrate Application of Seaweed Powder from Ulva ohnoi Triggers Physicochemical and Microbial Changes and Stimulates the Growth of Tomato Plants”, authored by Espinosa-Antòn and colleagues, deals with the evaluation of the potential effects of substrate changes induced by U. ohnoi application on the vegetative response of tomato plants under greenhouse conditions.
The manuscript is written with authority and contains truly interesting data. However, before it can be judged suitable for publication in Plants something need to be fixed.
(i) The title is really too long and complicated. It is unusual to find title of such complex scientific articles. Strongly suggest authors to make this section shorter. Normally, a single sentence summarizing the research purposes is recommended for the Title.
The title was change to: Application of seaweed powder generates changes in substrate and stimulates the growth of tomato plants.
(ii) In the affiliations section, authors should include acronyms for each component immediately after the respective e-mail address. These same acronyms should then be used in the contributions section.
The authors included acronyms for each component immediately after the respective e-mail address.
(iii) The abstract section is also too long. Normally this section is 200 words maximum. Incidentally, this is also a prerogative of the journal. I would suggest reading the guidelines carefully before the next submission.
The abstract section was improved changed to is 200 words maximum.
(iv) Some keywords should be changed. The utility of these terms is to facilitate the search of the article using common scientific search engines (PubMed, GoogleScholar, Scopus, etc.), which rely on the terms contained in title, abstract, and keywords. Consequently, using terms that are already in these sections as keywords is inappropriate. I strongly suggest that the repetitive keywords be changed before re-submission.
The keywords be change according suggestion of reviewer.
(v) The introduction contains a lack of information. In recent years, research is beginning to evaluate not only the effect of biostimulant application on plant physiology, but also on the quality of fruit that can be produced. This is a very important point, which has resulted in the shift of almost exclusive use of Arabidopsis in experimental trials. Tomato, finds an interesting application in these studies with biostimulants, precisely because the plant is capable of producing edible fruits. There are currently a limited number of scientific articles dealing with the potential effect of biostimulants on the nutritional, and nutraceutical properties of produced fruits. I strongly recommend that a paragraph in the introduction be introduced and relevant references included (10.3390/ijms232415911; 10.3390/agronomy12020428; 10.3390/plants11030348).
Line 47 to 55, the introduction section, was including information of the references (10.3390/ijms232415911; 10.3390/agronomy12020428; 10.3390/plants11030348). The authors agree with the reviewer comments, However, the authors only mentioned it briefly because, as a second phase of continuity of the work, we also evaluate the yield and the nutritional, and nutraceutical properties of tomato fruits treated with the same SP and SE-treatments (data not showed). We are preparing the writing of another article with this information.
(vi) Table lack in standard deviation. Did the authors performed more than one determination for this analysis?
(vii) Figure 1 and 3 also lacks standard deviations. Each individual point should have a bar reporting the standard deviation. If the authors feel that the resulting graph is too confusing, they should at least introduce information in the caption reporting that the coefficient of variation is always under a certain percentage. I also recommend reporting these data as mean and standard deviation in a supplementary table, since it is not apparent whether the data are actually statistically significant.
In tables 1 and 3 the standard deviation was included and figure 1 was improved, according to the reviewer's recommendation. Data were included in Supplementary Table 1.
(viii) table three should report an additional column with the P value.
In Table 3. Only showed the physicochemical characteristics of seaweed powder (SP) and seaweed extract (SE) from Ulva ohnoi, the chemical composition of the powder was not compared with the chemical composition of the extract. Therefore, P value is not included.
(x) SPAD detection of chlorophylls is certainly a very fast and low-cost method for these determinations. However, it turns out to be not very accurate. If the authors have the ability to perform quantifications spectrophotometrically or in HPLC of their degradation products it would be better.
It is true that the data evaluated with SPAD are not usually very precise, however previously in a previous study carried out by Hernandez-Herrera et al. 2022 and published in Agronomy if we analyzed these compounds by HPLC and SPAD and we observed that there was not much variation in the data. Therefore, it was decided to work with the SPAD data, which is a very fast and low-cost method for these determinations. Also, we did not have much biomass, which was used for mineral analysis.
(xi) Why is paragraph 2.4. called biochemical content? What do the authors mean by biochemical content? I don't think such a term is appropriate. Biochemistry studies biological macromolecules, how they function, and the processes in which they are involved. In this section I see none of that. The authors should rename this section as "Mineral composition and chlorophyll content" or something like that.
This section was changed to recommendation of reviewer.
(xii) letters related to ANOVA, when present in tables, should be position as superscript of the last number.
This was changed.
(xiii) In the affiliation section, authors should list their email addresses for each affiliation, along with their acronyms in parentheses. These acronyms should be the same as those used for the contributions section.
This was changed.
